# Predicting viral sensitivity to antibodies using genetic sequences and antibody similarities

**Kai S. Shimagaki**[1,2], **Gargi Kher**[1], **Rebecca M. Lynch**[3], **John P. Barton**[1,2*]

**1** Department of Computational and Systems Biology, University of Pittsburgh School of Medicine, Pittsburgh, Pennsylvania, United States of America, **2** Department of Physics and Astronomy, University of Pittsburgh, Pittsburgh, Pennsylvania, United States of America, **3** Department of Microbiology, Immunology and Tropical Medicine, School of Medicine and Health Sciences, George Washington University, United States of America

* jpbarton@pitt.edu

## Abstract

For genetically variable pathogens such as human immunodeficiency virus (HIV)-1, individual viral isolates can differ dramatically in their sensitivity to antibodies. The ability to predict which viruses will be sensitive and which will be resistant to a specific antibody could aid in the design of antibody therapies and help illuminate resistance evolution. Due to the enormous number of possible combinations, it is not possible to experimentally measure neutralization values for all pairs of viruses and antibodies. Here, we developed a simple and interpretable method called grouped neutralization learning (GNL) to predict neutralization values by leveraging viral genetic sequences and similarities in neutralization profiles between antibodies. The trained model is interpretable and can identify key mutations that impact viral sensitivity. Our method compares favorably to state-of-the-art approaches and is robust to model parameter assumptions. GNL can predict neutralization values for viral sequences without observed neutralization measurements, an important capability for assessing antibody coverage in populations whose viral diversity is genetically characterized. We also demonstrate that GNL can successfully transfer knowledge between independent data sets, allowing rapid estimates of viral sensitivity based on prior knowledge.

## Author summary

Antibodies play a central role in controlling viral infections, yet viral strains can differ widely in their sensitivity to a given antibody. Quantifying this sensitivity is critical for understanding immune escape, interpreting experimental and clinical studies, and designing effective antibody-based therapies. However, virus neutralization assays are labor-intensive and inherently sparse: only a small fraction of the vast space of possible virus-antibody combinations can be measured

**Data availability statement:** Sets of data and computer codes available in the GitHub repository: https://github.com/bartonlab/paper-ic50-prediction. The repository contains neutralization data sets and HIV-1 envelope genetic sequences. The most recent neutralization data and HIV viral sequences data sets are obtained from the CATNAP (http://hiv.lanl.gov/catnap) and LANL databases (https://www.hiv.lanl.gov), respectively.

**Funding:** This work was supported by the National Institute of Allergy and Infectious Diseases of the NIH award number R01AI152770 to RL. The funders had no role in study design, data collection and analysis, decision to publish, or preparation of the manuscript.

**Competing interests:** The authors have declared that no competing interests exist.

experimentally. Here, we present a computational framework, grouped neutralization learning (GNL), for predicting viral sensitivity across large virus-antibody panels using incomplete neutralization data and viral genetic sequences. The central idea of GNL is to share information across antibodies with similar neutralization profiles, allowing sparsely measured antibodies to benefit from related, more extensively characterized ones. Building on prior insights that neutralization data exhibit strong low-rank structure, we integrate antibody-level information sharing with sequence-based prediction and low-rank matrix refinement to produce robust and accurate neutralization predictions. Across large HIV-1 neutralization datasets, GNL performs favorably compared to state-of-the-art methods, particularly in when data is limited. Our model is interpretable, identifying specific viral mutations that strongly influence antibody sensitivity and recovering patterns consistent with known resistance mechanisms. Importantly, GNL can generate sensitivity estimates even for viral sequences without any measured neutralization data. Collectively, this work provides a scalable and interpretable framework for mapping virus-antibody interactions from partial data. While developed and validated using HIV-1, our approach is broadly applicable to other rapidly evolving pathogens for which genetic sequences and sparse neutralization measurements are available.

## Introduction

Viral pathogens such as HIV-1 exhibit extraordinary genetic diversity. Consequently, distinct viral strains can differ dramatically in their sensitivity to antibodies. Predicting whether a given virus is sensitive or resistant to a particular antibody has therefore become a key goal with multiple applications. Antibodies have been used as therapeutic treatments [1–3] and for passive immunization [4], which can fail when viruses harbor or develop resistance [5–7].

Passive immunization with broadly neutralizing antibodies (bnAbs) is a promising treatment to prevent infection or control viral progression, and its clinical relevance has been demonstrated in large-scale human studies such as antibody-mediated prevention (AMP) trials [8]. These efforts highlight the need for methods to systematically characterize how viral genetic variation affects antibody susceptibility or resistance. Computational methods that can predict viral sensitivity from genetic sequence data could offer a scalable way to assess antibody coverage, interpret experimental and clinical results, and inform the design of more effective antibody treatment interventions [9]. Such predictions would also reveal trends in antibody resistance over time, which could inform vaccine design for HIV-1 or other rapidly evolving pathogens.

Experiments alone are insufficient to fully characterize the virus-antibody sensitivity landscape. The space of functional virus sequences is so large that it is effectively unlimited, precluding exhaustive tests even for a single antibody. In response, a variety of computational methods have been developed to attempt to learn patterns of sensitivity and resistance from finitely sampled data [10].

For HIV-1, traditional machine learning approaches have used neutralization data coupled with HIV-1 envelope amino acid sequences to predict virus-antibody neutralization values. Support vector machines [11] and gradient boosting machines [12] have both been used to make binary predictions of sensitivity or resistance. Subsequent work combined additional features (e.g., geographic information for viral isolates) into a "super learning" framework to produce more precise predictions of neutralization values [13–15]. Methods have also incorporated virus-antibody structural data into a multi-layer perceptron model for resistance prediction, which predicts both binary sensitive/resistant states and IC50 values [16].

In recent years, deep learning methods have also been applied to this problem. One approach used recurrent neural networks to capture long-range dependencies in viral sequences [17], which was later extended to incorporate attention mechanisms that identify the most relevant amino acid positions for neutralization prediction [18]. Notably, the latter study incorporated antibody sequence data into the predictive model. Language models have also been applied for resistance prediction. Igiraneza and collaborators trained a model on HIV-1 envelope sequence embeddings using multi-task learning, where predictions are made across multiple antibody types simultaneously [19]. In competitive tests, deep learning methods often improved prediction accuracy, especially for antibodies with limited training data. However, these models are also more difficult to interpret, making the identification of resistance mutations challenging.

In parallel work, Einav and collaborators have taken a different approach [20–22]. They observed that the matrix of virus vs. antibody neutralization values exhibits a low-rank structure, with significant correlations across neutralization profiles. Einav et al. effectively learn a low-dimensional representation of the neutralization matrix to predict unobserved neutralization values. While this approach effectively captures antibody-virus similarities, it cannot predict resistance values for novel viruses because it relies on observed neutralization data rather than sequence or structural features.

A key challenge across these approaches is balancing predictive accuracy with generalizability. Sequence-based models risk overfitting due to their reliance on a large number of parameters, while methods based solely on low-rank approximation lack the ability to predict resistance patterns for novel viral sequences. Further advancements will likely require hybrid approaches that integrate both sequence-based and matrix-structure insights to enhance predictive robustness.

Here, we present a simple method for predicting viral sensitivity to antibodies based on finitely sampled neutralization data. We begin by quantifying similarities between antibodies based on their neutralization profiles across viral strains and clustering antibodies into distinct groups. Then, we project viral sequences into a low-dimensional space and train linear predictors of antibody sensitivity as a function of viral sequence features. Importantly, we use regularization to constrain the model such that the parameters for antibodies with similar neutralization profiles will also be similar. This allows us to leverage information across related antibodies to improve predictive power. Finally, we use a low-rank matrix approximation inspired by the work of Einav and collaborators [20–22] to further exploit antibody and virus correlations. Because of our focus on sharing information across groups of related antibodies and viruses, we refer to our approach as grouped neutralization learning (GNL).

We tested our approach using CATNAP, a large repository of HIV-1 virus-antibody neutralization data from in vitro TZM-bl assays [23]. The CATNAP database contains neutralization data for a diverse collection of antibodies, including but not limited to broadly neutralizing antibodies (bnAbs). Unless otherwise specified, our model is trained on and applied to this full set. Here we considered neutralization predictions for monoclonal antibodies only, and not for antibody mixtures. Overall, our method typically outperforms previous approaches that make the same types of predictions. GNL performs especially well for antibodies with limited observations, and in comparative tests, its predictions were notably more accurate for bnAbs directed toward the variable loops and the CD4 binding site of the HIV-1 surface protein. Furthermore, analyzing the trained weight parameters of the GNL model readily identifies key mutations that influence viral sensitivity.

Because GNL requires only a reference neutralization database (e.g., CATNAP) and viral genetic sequence data (here, HIV-1 Env), it can provide rapid sensitivity estimates without performing new neutralization assays for each virus-antibody pair. In many study settings, Env sequencing can be generated routinely from participant samples and used to screen

circulating or within-host variants. Computational approaches like GNL could therefore enable sequence-based sensitivity estimates to support tasks such as cohort stratification, prioritization of candidate bnAbs, or screening potential clinical trial participants when experimental neutralization testing is impractical. Overall, GNL improves our ability to predict virus-antibody neutralization values, which could be leveraged for further practical applications such as antibody therapy design [8].

## Results

### Grouped neutralization learning framework

One of the principal challenges of predicting virus-antibody neutralization is sparse data. Neutralization experiments are generally low throughput, and the space of possible virus and antibody sequences is enormous. We assume that our data consists of a set of antibody-virus neutralization values and corresponding virus sequences. The neutralization values can be arranged into an ($N_A \times N_V$)-dimensional matrix, $U$, where $N_A$ is the number of antibodies and $N_V$ the number of viruses, with the average neutralization value for antibody $\mu$ and virus $\alpha$ given by $u_{\mu\alpha}$ (for an example, see **Fig 1a**). Here, we will primarily use the half-maximal inhibitory concentration (IC50)—a standard summary statistic obtained by fitting antibody titration curves to estimate the antibody concentration required to achieve 50% neutralization—as our metric of virus "sensitivity" to antibodies. This value need not be defined for all antibody-virus pairs, and viruses may be included even if there are no neutralization values measured for them for any of the antibodies in the panel. Our goal is then to use the observed $u_{\mu\alpha}$ and virus sequences to predict the unobserved $u_{\mu\alpha}$. Here, we will assume that the neutralization values are quantified by IC50s on a logarithmic scale, but other metrics could also be used.

One of the main ideas of our approach is to leverage similarities between antibodies to improve predictive power. We also associate viral sequence features with antibody sensitivity or resistance, enabling predictions for novel sequences. Finally, we employ a low-rank approximation for the antibody-virus neutralization matrix, which makes use of common patterns in the matrix to refine the results. Below, we describe each of these steps (**S1 Fig**) and their rationale.

### Grouping similar antibodies

To identify antibodies with similar neutralization profiles, we calculate a similarity matrix between all antibody pairs based on the observed neutralization values (see **Fig 1b** for a visualization). The similarity $w_{\mu\nu}$ between antibodies $\mu$ and $\nu$ is defined as the angular similarity of their neutralization vectors against the same viruses (**S1 Text**). Thus, pairs of antibodies that have highly similar neutralization profiles against the same viruses will receive higher similarity scores.

We then apply a spectral clustering algorithm to group antibodies into a set of $K$ clusters [24]. Antibodies that exhibit similar neutralization activities across different viruses are grouped into the same groups. For the CATNAP data set, we used $K = 100$, though our results are robust to a range of values for $K$ between 50 and 200. Most clusters contain one or a few antibodies, 10–20 clusters contain 10–20 antibodies, and only a few contain around 100 antibodies.

### Extracting sequence features and learning neutralization values

Here, we used a simple approach to embed viral sequence data into a lower-dimensional space. We computed the eigenvalues and eigenvectors of the viral sequence covariance matrix and then projected each viral sequence onto the eigenvectors corresponding to the $d$ largest eigenvalues (**S1 Text**). Specifically, given a binary (one-hot) encoding of a viral sequence $\alpha$, $\boldsymbol{g}_\alpha \in \{0, 1\}^L$, we generate a projected sequence $\boldsymbol{x}_\alpha = \Xi\boldsymbol{g}_\alpha$. Here $\Xi \in \mathbb{R}^{d\times L}$ is a projection matrix, with $d \ll L$. This approach is statistically efficient, and it takes advantage of correlated patterns of mutational variation observed in the data, which can affect virus sensitivity to antibodies. For example, in the broad HIV-1 CATNAP data set, we observed that different viral subtypes tend to occupy distinct regions in the eigenspace due to shared, subtype-defining mutations (**S2 Fig**). While our main approach simply uses principal components, we also explored alternative dimensional reduction methods, including non-negative matrix factorization (NMF). However, the accuracy of our predictions changed

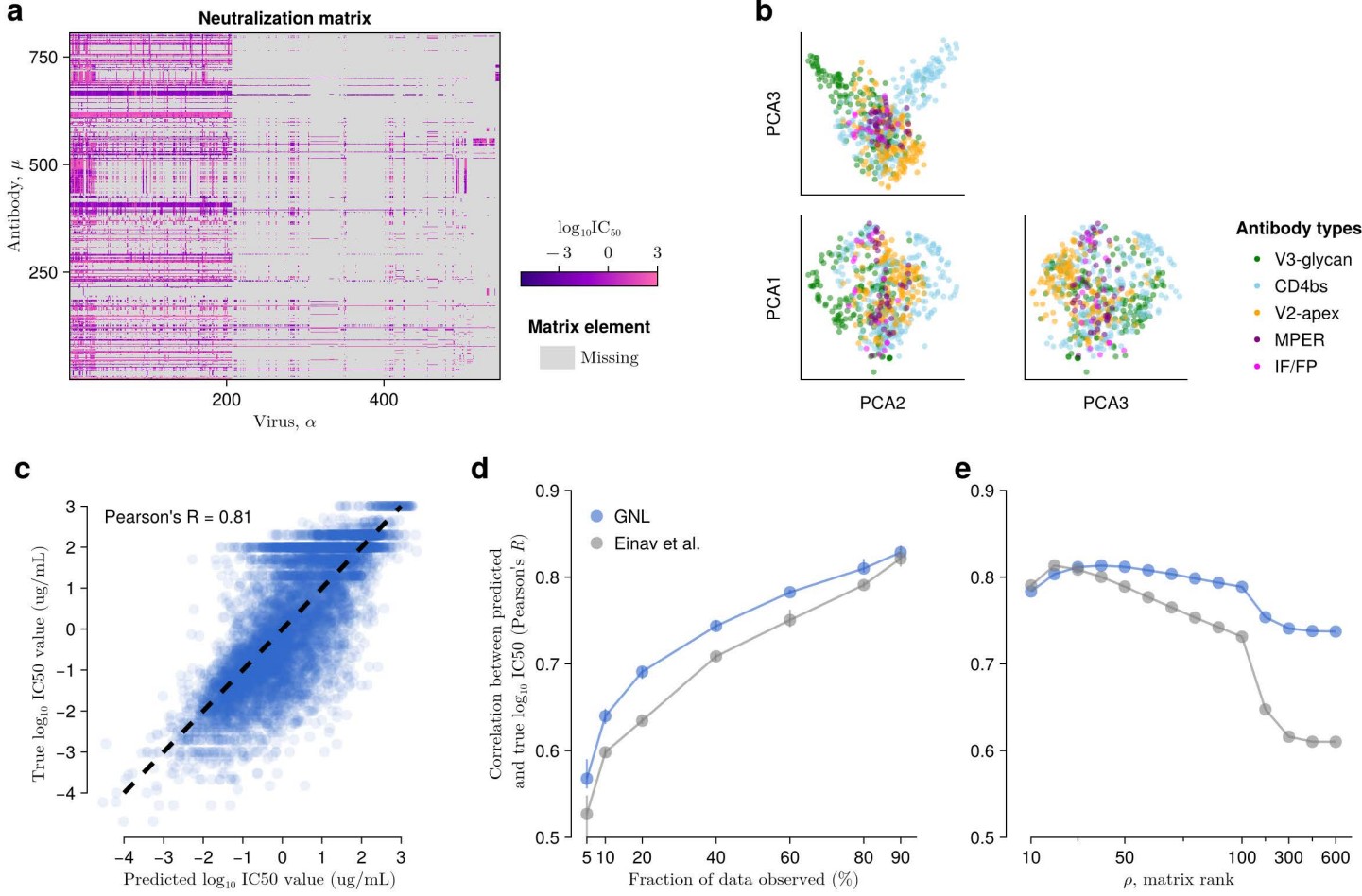

**Fig 1. GNL improves virus-antibody neutralization predictions. (a)** Visualization of HIV-1 neutralization data retrieved from CATNAP [23]. **(b)** Antibody similarity visualized by projection onto the top principal components of the similarity matrix $w_{\mu\nu}$. Antibodies are colored according to the part of the HIV-1 surface protein, Env, that they bind to. **(c)** Comparison of neutralization values predicted by GNL with true, withheld values. The rank of the low-rank approximation is selected as the smallest value for which the cumulative fraction of variance explained by the retained eigenmodes exceeds 95% (see **S1 Text**). Correlations between true and predicted neutralization values are highly significant, exhibiting a *p*-value of less than $10^{-288.8}$ across all conditions. **(d)** For both GNL and the Einav et al. approach, predictive power (measured by Pearson's $R$ between true and predicted neutralization values) improves along with the amount of data used for training. As in **Fig 1a**, the rank values are selected based on our optimization criteria. **(e)** Predictive power depends nonlinearly on the rank $\rho$ used in the low-rank matrix approximation for both GNL and Einav et al. for 80% of fraction of the data observed. However, our approach exhibits stable and generally improved performance over a wide range of $\rho$ values. Error bars represent 95% confidence intervals, obtained by randomly partitioning neutralization values into training or testing subsets.

little across different low-dimensional spaces as long as they provided statistically important features to distinguish viral sequences.

We used linear regression to learn the relationship between sequence features and neutralization values, inferring a separate family of regression models for each cluster of antibodies (**S1 Text**). For each cluster, we set $d$ to be proportional to the number of unique viruses with measured neutralization values for antibodies in the cluster, up to a maximum of 100. We used the antibody similarity measures defined above, $w_{\mu\nu}$, to penalize differences between the linear regression parameters $\boldsymbol{\theta}_\mu$ and $\boldsymbol{\theta}_\nu$ for each pair of antibodies $\mu$ and $\nu$ within the same cluster. In this way, similar regression parameters will be learned for antibodies with similar neutralization profiles (i.e., $w_{\mu\nu} \sim 1$). In the limit that $w_{\mu\nu} = 0$, the regression

parameters for antibodies $\mu$ and $\nu$ become independent. Overall, this approach allows the information from virus-antibody neutralization data to be shared among similar antibodies.

## Low-rank approximation for the neutralization matrix

Einav and collaborators found that typical virus-antibody neutralization matrices have a "low-rank" structure [20, 22, 21]. In other words, the neutralization profiles for each virus and antibody are not completely unique; virus-antibody neutralization scores tend to be highly correlated across antibodies (for closely-related viruses) and across viruses (for antibodies that bind similar epitopes in a similar way). Thus, the neutralization matrix can be well-characterized by a number of modes much smaller than its naive dimensions. Our approach makes use of the same insight.

After inferring linear regression parameters for each antibody $\boldsymbol{\theta}_\mu$, we used the regression model to predict the neutralization value $u_{\mu\alpha}$ for each virus-antibody pair that was not measured in the data set. We then refined this matrix by applying a low-rank approximation. Specifically, we used singular value decomposition (SVD) to factorize the preliminary $U$ matrix, consisting of the measured $u_{\mu\alpha}$ with the unobserved entries filled by the regression model. We then selected the top $\rho$ singular values and basis vectors to approximate the matrix. As in prior work [20], we selected the rank $\rho$ to be the smallest rank such that the cumulative variance explained (equal to the sum of squared singular values) is greater than 95%. Our final predictions for the unobserved $u_{\mu\alpha}$ are then given by the entries of this low-rank approximation of the completed $U$ matrix after regression.

In the previous work, Einav et al. used robust principal component analysis or nuclear norm minimization to develop a low-rank approximation of the neutralization matrix [20]. We also tested these methods, but in our analyses, we found that there was little difference between them and the simpler SVD factorization. Consistent with Einav et al., we also observed that the spectrum of singular values could reveal information about the neutralization matrix. We found that the fraction of variance explained scales roughly as a power law for intermediate eigenmodes, ranging from around rank 20–100 (**S3 Fig**).

## Neutralization and virus genetic data

In this study, we consider two types of neutralization matrices derived from monoclonal anti-HIV-1 antibodies and HIV-1 strains. The first is a neutralization matrix obtained from cross-sectional studies in the CATNAP database [23] provided by Los Alamos National Laboratory (LANL) [25]. To ensure data quality, we first removed viruses and antibodies that had few associated neutralization values (<10 entries for viruses), and then antibodies that do not have any entries. We also removed viruses that had not been sequenced.

After filtering, we obtained a data set with $N_A = 806$ antibodies and $N_V = 1{,}147$ viruses, including 81,193 observed neutralization values. As indicated above, neutralization data in large data sets such as CATNAP are sparse: the measured neutralization values correspond to 4.5% of the total matrix entries.

The second data set comes from an HIV-1 intrahost longitudinal study, which tracked the development of bnAbs (CH235 and CH103) in one donor, CH505 (ref. [26,27]). We obtained virus-antibody neutralization values from Gao et al. [27] and corresponding HIV-1 Env sequences from LANL's HIV sequence database [25]. This matrix includes $N_A = 22$ antibodies and $N_V = 109$ virus sequences, with 1,723 measured neutralization values. In contrast to the CATNAP database, 71.9% of the possible neutralization matrix entries were measured in this data set. More details on the formatting of the neutralization matrices, the processing of genetic sequences, data sources, and the parameters of the models are provided in **S1 Text**, **S1 Table** and **S2 Table**.

## Overall GNL performance on CATNAP data

To test the performance of the GNL method, we masked a random fraction of virus-antibody neutralization values in the CATNAP data set. We then used GNL to predict the masked neutralization values and compared our predictions against

 

the true measured values. First, we masked entries of $U$ uniformly at random (i.e., each observed $u_{\mu\alpha}$ value was equally likely to be masked). **Fig 1c** shows a typical comparison between the true and predicted neutralization (log IC50) values when 20% of the data is masked, which are strongly correlated. In this example, there are 780 antibodies and 1,097 viruses in the withheld data.

We also compared our approach versus that of Einav et al. (ref. [20]) as we varied the fraction of the data used for training and validation, and the rank $\rho$ used in the low-rank matrix approximation. To measure typical performance, we repeated each test 10 times for each data fraction and $\rho$ value. Overall, we observed that the correlation between true and predicted log IC50 values was typically higher for GNL than for the alternative (Fig 1d, e). Comparatively, the results for GNL were especially better when the neutralization data were sparser. We also observed that GNL performance was more robust to changes in the rank $\rho$, with a wider plateau near peak performance. This robustness is especially important for practical applications, where the true neutralization values are unknown, because it implies that strong predictions can be obtained even without fine-tuning parameters.

### Determinants of prediction accuracy

Next, we aimed to identify conditions that make predicting virus-antibody neutralization values more or less difficult. We focused on two key factors in the neutralization data: the diversity of neutralization values and the amount of data available (Fig 2a-b; see also **S4 Fig**). Intuitively, antibodies with more observed neutralization values provide more information for training, potentially enabling higher accuracy. Antibodies with a broad range of neutralization values (i.e., ones with substantially different IC50 values for different virus strains) may also be more difficult to model than those with flatter neutralization profiles.

Because some subsets of the data contained only a small number of virus-antibody pairs, we used the mean squared error (MSE) between predicted and observed neutralization values to quantify accuracy in these comparisons. With few data points, correlation metrics such as Pearson's $R$ may not fully capture the correspondence between predictions and data. This is especially true for neutralization values with little variation, where predictions may be quite accurate on an absolute scale even if they are not strictly ordered correctly. For example, antibodies VRC34-UCA, VRC34.05, vFP5.01, A12V163-a.02, DH650.8, and m66 have more than 200 neutralization observations, but the variance in measured log IC50 values for each is less than 0.1.

For reference, we compared our results against a simple version of our model where we treat each antibody independently. This independent neutralization learning (INL) model is equivalent to simple linear regression to predict neutralization values for each antibody. Overall, we found that MSE values tend to decrease as the amount of training data increases and as the variance in the training data decreases (Fig 2c-e). Errors for GNL were always smaller than those for INL, demonstrating the value of shared information across antibodies with similar neutralization profiles.

We grouped antibodies into five classes according to the standard deviations of their neutralization values. Fig 2f-g shows that MSE increases as the diversity of log IC50 values increases. Consistent with analyses in **Fig 1**, we set the fraction of data observed as 80%. More generally, when large fractions of the observed data are used for training, the difference in MSE values between the GNL and INL methods becomes more pronounced when the diversity of neutralization measurements is higher (Fig 2h), a setting in which learning neutralization values is more challenging.

We further partitioned antibodies based on the number of neutralization measurements used for training (Fig 2i-k). Naturally, MSE values tend to be larger for antibodies with fewer training data. However, the advantage of GNL over INL is especially prominent in the cases where data is most restricted, including a small fraction of the data observed, smaller training data sets, and less variation in the training data (Fig 2h, k). This suggests that the grouping mechanism in GNL, which allows predictive information to be shared across similar antibodies, is especially helpful in learning genotype-to-neutralization mappings with sparse data.

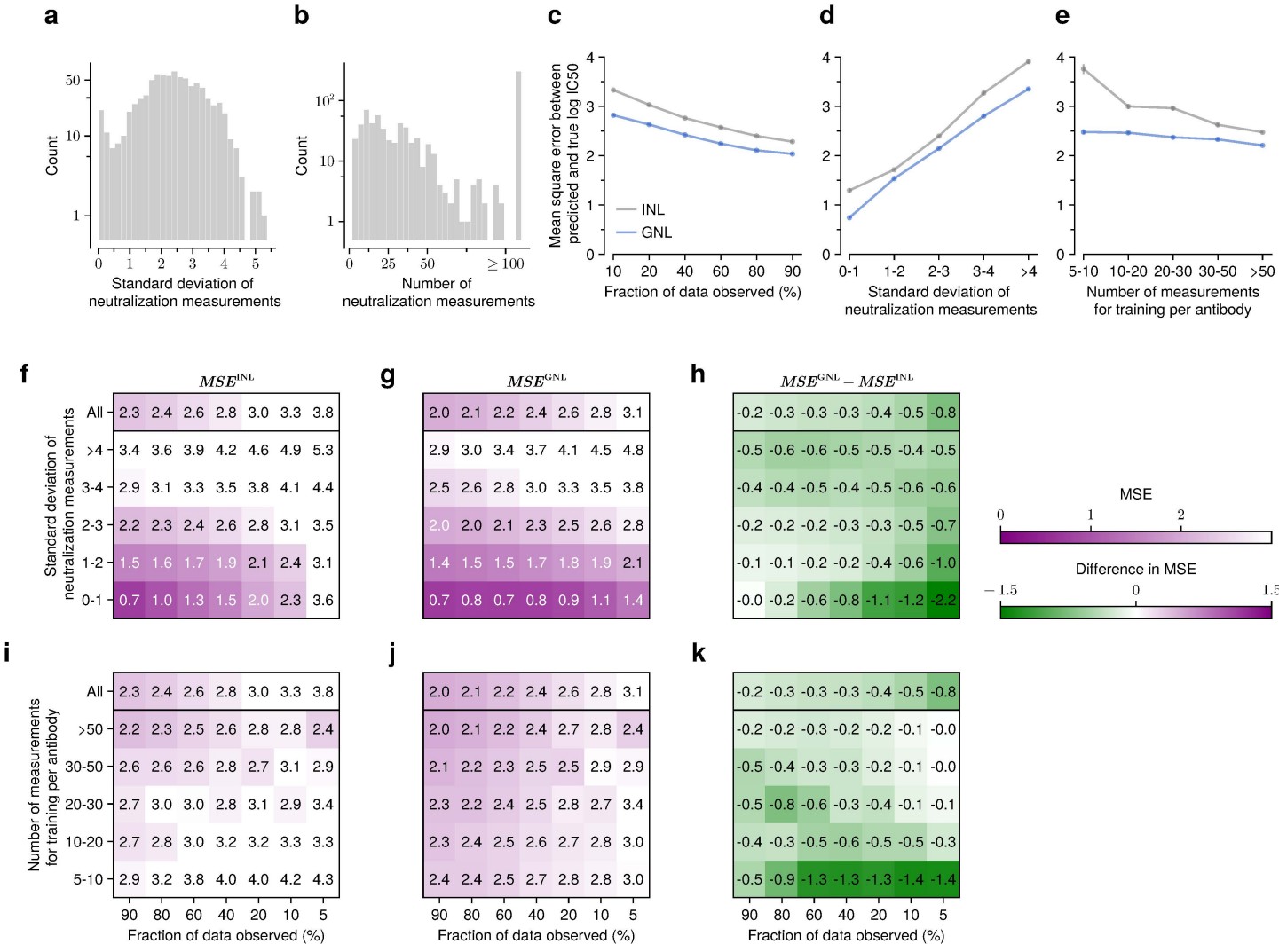

**Fig 2. GNL improves predictions when neutralization measurements are widely divergent or when data is sparse.** Overall distribution of the standard deviation of the neutralization measurements per antibody **(a)**, and their number of neutralization measurements **(b)**. As the fraction of observed data decreases, accuracy drops (i.e., MSE increases); however, GNL consistently shows lower MSE, especially at smaller data fractions **(c)**. GNL also performs better for antibodies with higher diversity, as measured by standard deviation **(d)**, or with fewer observed measurements used for training **(e)**. Greater diversity in neutralization measurements increases MSE across all withholding levels for both INL and GNL **(f-g)**. However, GNL better mitigates this accuracy drop **(h)**. Similarly, we observed that fewer training measurements lead to higher MSE across all withholding levels **(i-j)**. However, the GNL method maintains lower MSE with fewer observations **(k)**. S4 Fig presents the same analysis including the Einav et al. method.

## Integration of alignment-free sequence features

HIV-1 Env hypervariable regions involve frequent mutations, insertions, and deletions, which can make sequence alignment challenging in these regions. A prior study incorporated alignment-free features to characterize patterns associated with viral sensitivity to bnAbs [28]. In this study, we also explored the predictive power of models that combine alignment-free features along with the sequence-based features described above.

We retrieved alignment-free features, such as loop length, net charge, and the number of potential N-linked glycosylation sites (PNGSs), within the five variable regions (V1–V5), totaling about 30 additional features, from the LANL

database. In addition to these alignment-free features, we used sequence alignments that include an additional letter 'O' to represent PNGSs, alongside the standard 20 amino acid letters. These additional features are incorporated into the GNL framework (see S1 Text for details). To evaluate neutralization prediction accuracy from multiple perspectives, we considered several statistical metrics, including Pearson's $R$, Spearman's $\rho$, regression slope, concordance correlation coefficient, and MSE.

Here, we found that the effect of including alignment-free features and the PNGS symbol in the model was minor, and we observed no significant difference in performance with or without these additional features in this setting (Fig 3). As discussed in the next section, we also examined the influence of these additional features on predicting other viral sensitivity measures such as IC80 and Hill slope. While their overall effect remains small, we observed slight improvements in accuracy for certain cases, such as the Hill slope (S5 Fig, S6 Fig and S7 Fig). Therefore, alignment-free features could help identify patterns in viral sensitivity and improve interpretability. However, alignment-based features were sufficient to learn IC50 values in this example.

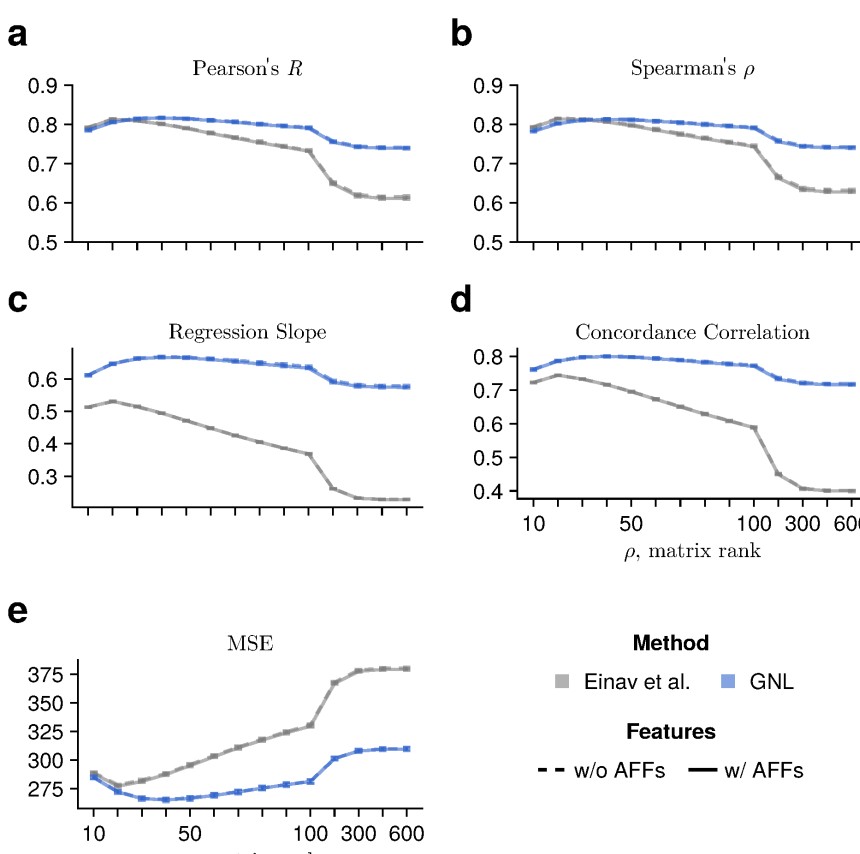

**Fig 3. Accuracy of IC50 predictions assessed by multiple statistical metrics.** We compared the performance of models that learn the relationship between neutralization and genetic sequences using only sequence alignments with the standard 20 amino acids and gaps (dashed lines) to models that incorporate both sequence alignments and alignment-free features (AFFs). Specifically, we used loop length, net charge, and the number of PNGSs in variable loop regions (solid lines) for the AFFs. To assess accuracy, we employed multiple metrics: **(a)** Pearson's $R$, **(b)** Spearman's $\rho$, **(c)** regression slope, **(d)** concordance correlation coefficient, and **(e)** MSE. Error bars represent 95% confidence intervals, estimated by randomly resampling the training and evaluation sets ten times. The concordance correlation coefficient (panel **d**) is computed as $r_c = 2r\sigma_x\sigma_y/(\sigma_x^2 + \sigma_y^2 + (\mu_x - \mu_y)^2)$ where $r$, $\mu_x$, and $\sigma_x^2$ denote the Pearson correlation coefficient, mean, and variance of one dataset ($x$-axis), respectively. Overall, models that include alignment-free features achieve virtually the same prediction accuracy across a wide range of matrix ranks and across all evaluated metrics.

## Predicting viral sensitivity metrics beyond IC50

IC50 is just one of multiple possible metrics that can be used to assess viral sensitivity to antibodies. IC80 (the 80% inhibitory concentration) estimates the antibody dose required to achieve a greater reduction in viral replication than IC50. The Hill slope $h$ describes how steeply neutralization changes as antibody concentration varies; by definition, it quantifies how sensitive neutralization titers are to changes in dosage. We characterized the Hill slope by IC50 and IC80 values, such that

$$h = \frac{\log(4)}{\log(\text{IC80}/\text{IC50})} .$$

(1)

Another important metric is the instantaneous inhibitory potential (IIP) [29,30]. Mathematically, the IIP at antibody concentration $c$ is defined as

$$\text{IIP}(c) = \log_{10}\left(1 + c/\text{IC50}\right)^h .$$

(2)

By definition, IIP quantifies the order of magnitude of the viral population that is neutralized (e.g., IIP = 1 corresponds to 90% neutralization, IIP = 2 corresponds to 99% neutralization, and so on).

To assess the robustness of the proposed prediction method, we explored predictions of IC80, Hill slope, and IIP values. Although IC50 has the most extensive collection of observations in the CATNAP database, IC80 values are also available, and a subset of measurements includes both IC50 and IC80. From these data, we estimated Hill slope $h$ and IIP values using the expressions above. We evaluated IIP at a fixed antibody concentration of $c = 1000\,\mu$g/mL throughout this study. Because IC50 and IC80 values are often very close, accurately estimating the Hill slope can be challenging. Therefore, we considered only Hill slope values below a threshold, specifically setting a maximum Hill slope of $h_{max}$ (corresponding to a very steep sigmoidal curve), and treated only values below this threshold as reliable observations.

Overall, we observed consistent trends in the relationship between observed fraction and accuracy (Fig 4), the relationship between prediction accuracy and matrix rank, with reduced matrix rank generally improving prediction accuracy (S5 Fig). Our method is robust to the choice of matrix rank, and Pearson's $R$ exceeds 0.8 across a wide range of rank values when 80% of the data are observed. However, we note that prediction accuracy for Hill slope and IIP values is relatively insensitive to matrix rank (S5 Fig, S6 Fig and S7 Fig), and that GNL's Pearson and Spearman correlation coefficients are comparable to or slightly lower than those obtained using the method of Einav et al. In contrast, for the regression slope and concordance correlation coefficient, the GNL method achieves higher performance than the other methods across all scenarios (S8 Fig and S9 Fig). Therefore, these results indicate that learning the relationship between genotypes and Hill slope is more challenging than learning IC50 or IC80 independently. This may be due to statistical uncertainty in estimating the Hill slope from IC50 and IC80 values, which can also propagate into IIP estimates.

## Interpretability of model parameters

HIV viral sensitivity to antibodies is strongly influenced by amino acid variation in the Env protein, and certain mutations, such as those leading to glycan loss or shifts (e.g., at N332), can substantially impact sensitivity. To gain a better understanding of the genotype–sensitivity relationship, we analyzed which mutations most strongly contribute to the predicted neutralization values.

We first obtained the trained model parameters learned from all observed IC50 values using the GNL method, which we consider to provide the most accurate model. We then calculated effective mutation-specific weight parameters, derived from the trained model parameters and the transformation mapping one-hot encoded sequences to projected variables (S1 Text). These effective weights correspond to the contributions of individual mutations to the predicted $\log(\text{IC50})$ values. Using these weights, we analyzed mutation contributions for specific classes of antibodies.

PLOS Computational Biology

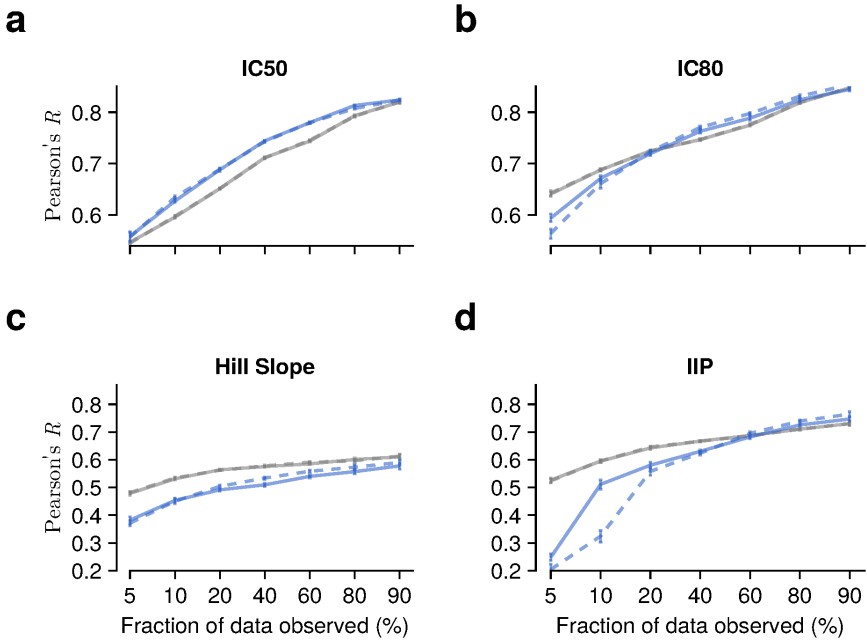

**Fig 4. Prediction of viral sensitivity measures beyond IC50. Pearson's *R* values as a function of the fraction of observed data are shown for (a) IC50, (b) IC80, (c) Hill slope, and (d) instantaneous inhibitory potential (IIP), using the optimized rank values (S1 Text).** Overall, increasing the fraction of observed data improves prediction accuracy across all metrics. However, for Hill slope and IIP, Pearson's *R* values obtained using the GNL method are lower than those from the method of Einav et al., whereas GNL achieves higher regression slope, concordance correlation coefficients, and MSE values (S8 Fig and S9 Fig). This suggests that while GNL captures the overall scale and agreement of predictions, rank-based correlation is more difficult to achieve for these measures. We speculate that predicting Hill slope involves a more complex relationship with genotypes, likely because Hill slope is estimated from a limited number of IC50 and IC80 pairs, which introduces additional statistical uncertainty.

This analysis reveals the effective contributions of individual mutations across sites and amino acids, averaged over multiple antibodies within each antibody class (**Fig 5a**). V3-glycan antibodies are known to depend on the N332 glycan; therefore, the presence of a PNGS ('O') at position 332 is expected to decrease IC50, while loss or alteration of this glycan can lead to viral escape and increased IC50 values. Consistent with this expectation, for V3-glycan antibodies we observed that the contribution of mutation O332 is significantly negative; this mutation ranks 9.5 on average, corresponding to the top 0.1% of contribution values across all 110 V3-glycan antibodies ($p = 10^{-105.6}$). Mutations N332 and T332 are significantly positive contributions, ranking 411.5 (5.0%) and 259.0 (3.1%), respectively when averaged over antibodies ($p = 10^{-26.4}$ and $p = 10^{-31.9}$, respectively; **Fig 5b**). Here, *p*-values are based on Fisher's exact test, and the top 1% of mutations are selected according to the absolute values of their effective weights (**S1 Text**). Similarly, mutations S334 and O334 exhibit significant positive and negative contributions, respectively, ranking 8.5 (top 0.1%) and 42.0 (0.5%) in average contriubtion rank across all V3-glycan antibodies ($p = 10^{-116.9}$ and $p = 10^{-80.3}$; **Fig 5c**). We also examined effective weight values for other antibody classes. For example, for V2 apex antibodies, negatively charged mutations are known to confer resistance. Consistent with this observation, our analysis shows that the positively charged mutation K171 decreases IC50 values and ranks 83 (top 1.0%), with a corresponding *p*-value of $10^{-66.1}$. (**S10 Fig**).

Beyond these specific mutations, we systematically examined mutations that exhibit significantly large contributions to the predicted log IC50 values and are shared across antibodies within the same class. Using these criteria, we selected 7, 5, 11, 8, and 6 mutations for the V3-glycan, CD4bs, V2-apex, MPER, and Interface/FP antibody classes, respectively (**S3 Table**). Among the 37 selected mutations, 23 are associated with altered viral sensitivity, and all mutations identified in the V3-glycan class affect glycan-related features (**S3 Table**). The majority of these mutations exhibit consistently positive or

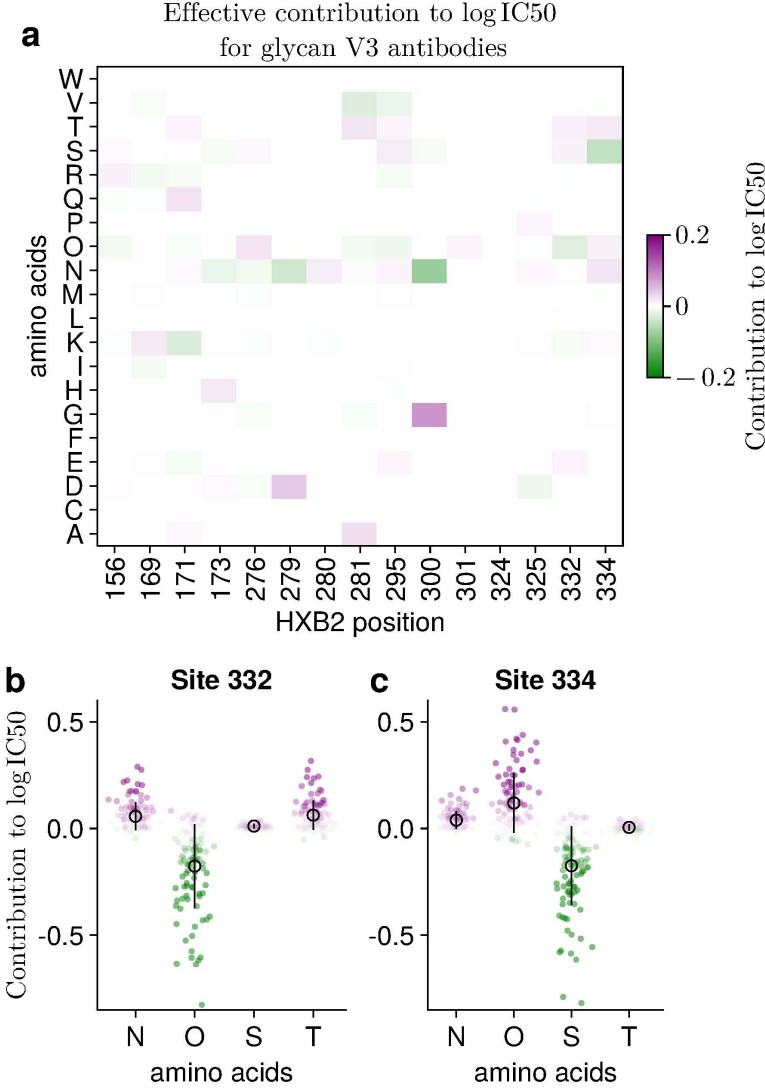

**Fig 5. GNL's model parameters reveal key mutations that impact viral sensitivity.** We show the contributions of individual mutations to the predicted log(IC50) values, which indicate how much the predicted log(IC50) changes in the presence of each mutation. These estimated contribution values are averaged over all V3-glycan antibodies, comprising 110 antibodies. The selected sites include residues that can directly contact or indirectly influence binding affinity with V3-glycan antibodies [31–36]. Overall, mutations involving 'O', 'N', 'T', and 'S' tend to contribute significantly to the predicted IC50 values **(a)**. Here 'O' indicates PNGS. We further show the contributions to log(IC50) for individual mutations at sites 332 and 334, which are frequently associated with glycan involvement **(b, c)**. Mean contribution values across all antibodies are indicated by open circles.

negative contributions to log IC50, corresponding to increased resistance or sensitivity to antibody neutralization. In contrast, mutations associated with the V2-apex class often display context-dependent, or dual, effects (**S11 Fig**).

## Comparison against alternative approaches for virus-antibody neutralization prediction

Alongside low-rank approximation methods, there are other families of neutralization prediction approaches, including ones based on machine learning [11–13,16,18]. One widely used tool is SLAPNAP (Super Learner Prediction of NAb Panels) [13]. SLAPNAP has been applied to several studies and predicts neutralization values against bnAbs using viral

envelope protein sequence features [14,15]. The underlying models include random forests and boosted regression trees with $L_1$ regularization. For combinations of bnAbs (a setting that we do not consider in this work), SLAPNAP uses the Bliss-Hill model [37], which estimates single bnAb neutralization curves via a Hill function and combines their effects based on Bliss independence.

To compare SLAPNAP's performance with the GNL method as well as the Einav et al. method [20], we first replicated their training settings. In particular, we trained each model using $V$-fold cross-validation. In this setup, the data is divided into $V$ parts: $V - 1$ are used for training, and the remaining part is used for validation. This is repeated $V$ times. We set $V = 5$, following prior work [13]. To ensure a fair comparison, we used the same neutralization data set (from CATNAP, as of July 23, 2021) previously tested for SLAPNAP. Following ref. [13], we used a subset of bnAbs that bind to different Env epitopes to compare the performance of GNL, SLAPNAP, and the Einav et al. method. Compared to the full CATNAP data set, the selected antibodies often had a large number of neutralization measurements (>200, for comparison see **Fig 1b**).

**Fig 6** reports CV-R values for GNL, SLAPNAP, and Einav et al. methods on this data set, showing that GNL performs well across a wide range of antibodies. Across all antibodies, GNL achieved an average CV-R of 0.60, compared to 0.51 for SLAPNAP and 0.43 for the Einav et al. approach (**S4 Table**). For antibodies DH270.5, DH270.6, and VRC38.01, SLAPNAP was unable to produce a CV-R value. These antibodies have fewer neutralization measurements than other antibodies in this set, and thus the lack of a CV-R result may be due to a data restriction in SLAPNAP. To ensure a fair comparison across methods, we thus excluded antibodies for which SLAPNAP did not produce CV-R values. We also reported the Spearman's correlation value based on cross-validation in the Supplementary section (**S12 Fig**).

## Zero-shot prediction of neutralization values for novel viral sequences

Next, we tested the ability of GNL to perform "zero-shot" predictions of virus-antibody neutralization values for virus sequences that never appeared in training data. Predicting neutralization values for novel sequences is challenging because it requires learning generalizable relationships between virus properties (e.g., sequence, structure) and resistance, rather than simply relying on observed neutralization patterns. Matrix completion methods that depend exclusively on observed virus-antibody pairs cannot make predictions for entirely new viral sequences. To evaluate GNL's zero-shot capabilities, we considered two experimental scenarios using longitudinal data from a single HIV-1 infected donor, CH505, and cross-data set predictions between independent neutralization studies.

As a first example, we considered longitudinal neutralization data from donor CH505, who was monitored over five years and developed bnAb lineages, CH103 and CH235, binding to CD4bs (Refs. [26,27]; see also **S5 Table**). The CH505 data set includes neutralization values for $N_A = 22$ antibodies tested against $N_V = 109$ viruses over six time points [26,27]. To validate our approach, we first confirmed that GNL could accurately predict randomly withheld neutralization values, achieving strong correlation ($R = 0.89$) when 80% of the data was observed (**S13b Fig**, see also **S1 Text** for the rank optimization scheme). We then examined two temporal prediction scenarios: predicting neutralization values for past viral sequences using models trained on later time points, and predicting future viral sequences using models trained on earlier data.

For temporal predictions, we split the neutralization data based on collection time, using early weeks post-infection (28, 96, and 140 weeks) and later weeks (210, 371, and 546 weeks) as separate training and validation sets. When predicting antibody-virus neutralization values for early viruses based on late infection data, GNL achieved Pearson's $R$-values of approximately 0.8 across all time points (**Fig 7c**). For the more challenging task of predicting neutralization values for future viral sequences based on prior (early infection) data, GNL maintained good predictive accuracy (Pearson's $R$ ranging from 0.7 to 0.55) despite the continued accumulation of mutations (**Fig 7b**). This reflects the ability of GNL to learn generalizable maps from viral sequence to neutralization.

In a second test, we evaluated whether GNL could predict neutralization values for an individual host using only data from the broader CATNAP database. We trained models exclusively on CATNAP data and predicted neutralization

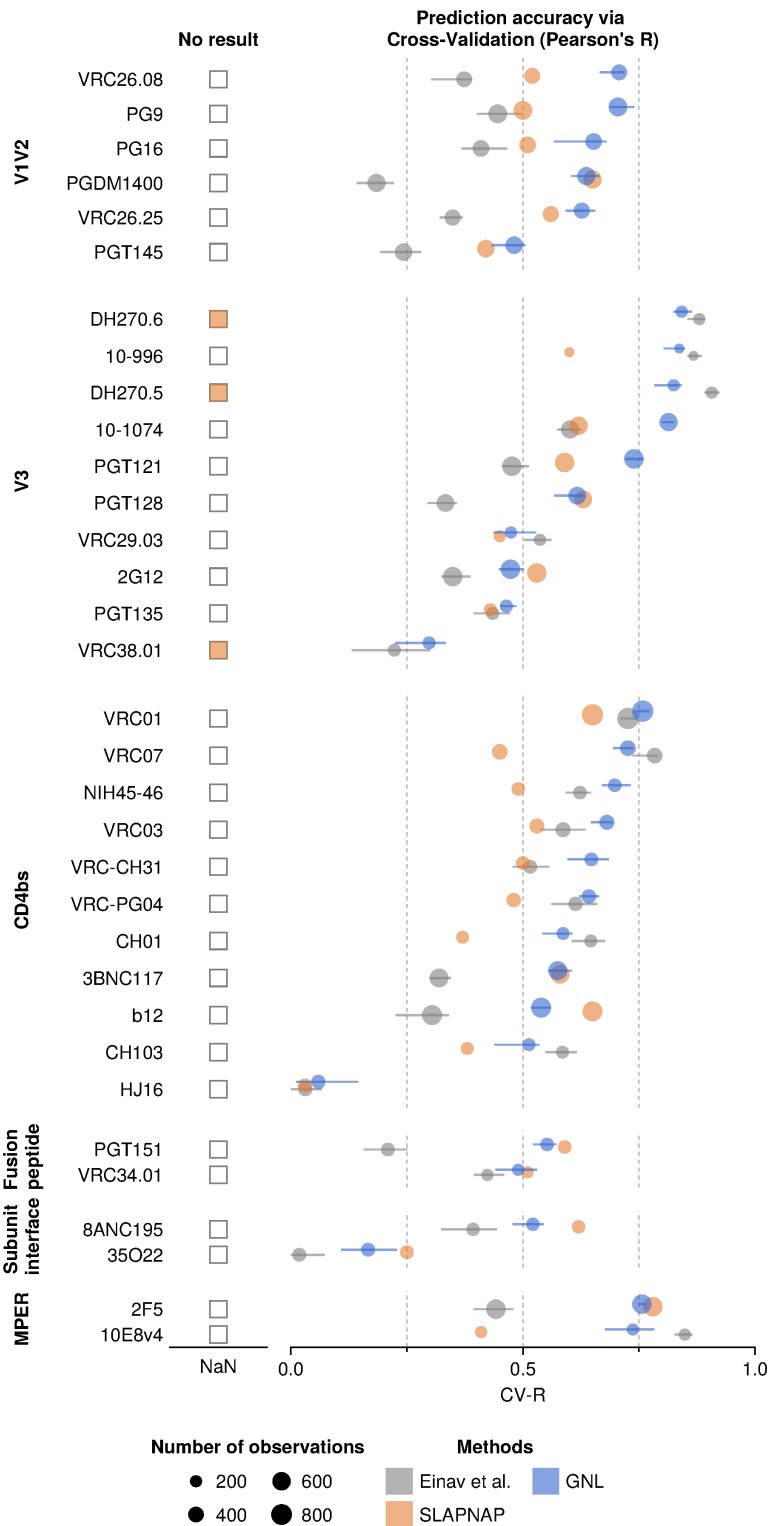

**Fig 6. Performance comparison of GNL and existing methods on CATNAP data.** Cross-validation Pearson's *R* (CV-R) for neutralization predictions of bnAbs across a spectrum of antibodies with GNL, SLAPNAP [13], and Einav et al. For a few antibodies (DH270.5/6 and VRC38.01), SLAPNAP produced no output. The average CV-R of GNL across all antibodies (0.60) exceeded SLAPNAP (0.51) and Einav et al. (0.46) (S4 Table). GNL performance was especially strong for antibodies binding the variable loops (V1/V2 and V3) and the CD4 binding site. Error bars represent 95% confidence intervals based on a random selection of viruses used for cross-validation.

values for the CH505 individual host data set, focusing on the two antibodies that are present in both the CATNAP and CH505 data sets (**Fig 8a**). Since the CATNAP database and individual host data are largely disjoint, with none of the CH505 virus sequences present in CATNAP, this represents a stringent test of zero-shot predictive power. Here, GNL achieved meaningful predictive accuracy with a Pearson's correlation of $R = 0.55$ between predicted and true neutralization values (**Fig 8c**). In comparison, predictions based on matrix completion are highly limited due to the absence of the specific test virus sequences in the training data (**Fig 8b**). These results show that GNL can successfully transfer neutralization patterns from cross-sectional studies to predict virus-antibody interactions in independent data sets.

## Discussion

Quantifying the efficiency of drugs or antibodies against pathogens is a fundamental task in biology. However, experimental neutralization measurements can be resource-intensive, and evaluating all possible combinations of treatments and pathogens is infeasible. We developed grouped neutralization learning (GNL), a simple method for predicting viral sensitivity to antibodies based on finitely sampled neutralization data. GNL employs three key ideas to boost predictive power: sharing information among antibodies with similar neutralization profiles, mapping from viral sequence to neutralization using regularized linear regression, and refining predictions using a low-rank matrix approximation, inspired by refs. [20,22]. In contrast with previous approaches based on matrix completion, GNL can use sequence features to predict neutralization values even for viruses with no prior neutralization data. We used GNL to predict half-maximum inhibitory concentration (IC50) values, but in principle the same approach could be applied to other similar metrics.

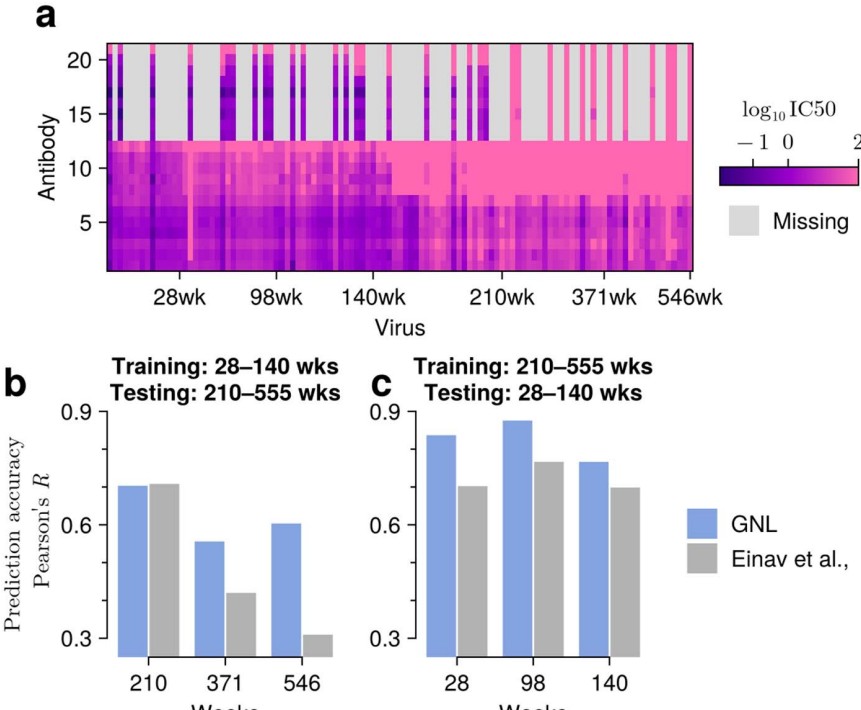

**Fig 7. GNL accurately predicts neutralization values for viral sequences across evolutionary time. a**, Visualization of the neutralization matrix, consisting of $N_A = 22$ antibodies and over $N_V = 109$ viral sequences sampled across six time points. **b**, Prediction of the neutralization values for past viral sequences. Neutralization values were predicted for sequences sampled at or before 140 weeks, using a model trained only on data collected after 140 weeks. **c**, Prediction of the neutralization activities for future viral sequences. The model was trained on values observed up to 140 weeks, and predictions were tested on sequences sampled after 210 weeks. GNL maintains accuracy even as viral sequences diverge from training data over time.

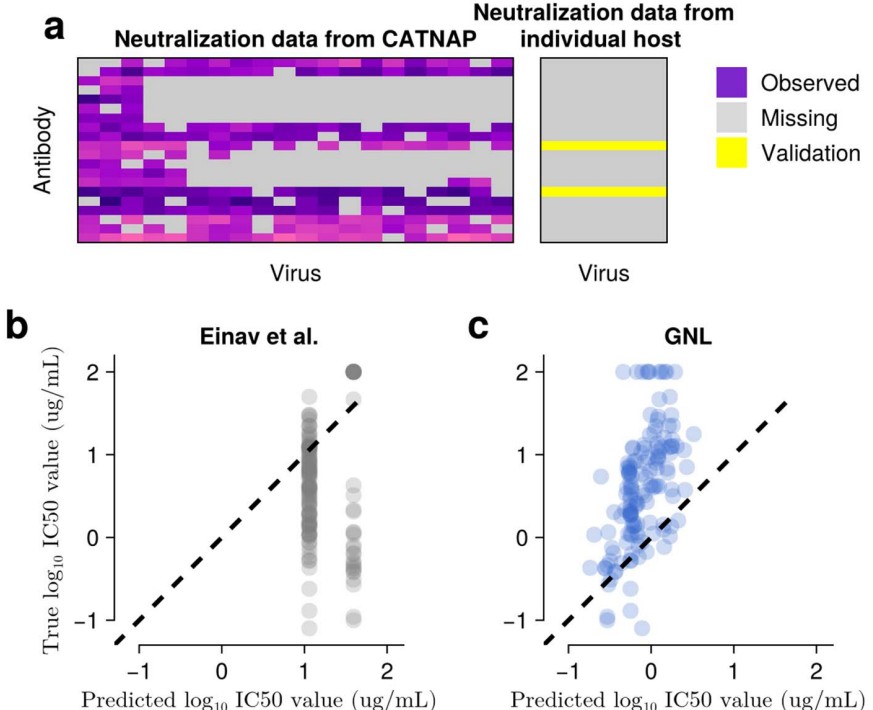

**Fig 8. Neutralization values can be predicted for novel viruses using cross-data set transfer learning. a**, Schematic showing a subset of the CAT-NAP database (left) and the individual host (CH505) neutralization matrix to be predicted (right). CATNAP contains $N_A$ = 806 antibodies and $N_V$ = 1,147 viruses, while the CH505 matrix includes 22 antibodies and 109 viral strains. Only two antibodies are shared between the data sets. **b**, Comparison of true neutralization values from CH505 and predicted values using the Einav et al. method [20], which can only predict mean values from the training data. **c**, GNL achieves meaningful predictive accuracy for the same task. GNL's (Einav et al.'s) Pearson's $R$, Spearman's $\rho$, linear regression slope, MSE, $p$– values are 0.55 (–0.02), 0.57 (–0.11), 1.53 (–0.07), 11.05 (11.28), and $10^{-11.8}$ ($10^{-0.1}$), respectively. As indicated by the regression slope, the predicted values are systematically smaller than the true ones in this example.

In competitive tests, our method compares favorably with the current state-of-the-art across a variety of scenarios. Our approach particularly excels in challenging conditions where data is sparse, using shared information from antibodies with similar neutralization profiles to enhance predictions (**Fig 2**). GNL is also especially robust to the choice of the matrix rank used in the low-rank matrix approximation step (**Fig 1**). This robustness is particularly valuable for practical applications where the rank must be chosen heuristically.

Our ability to predict neutralization values for novel viral sequences enables several important biological applications. In a test on longitudinal data from a single donor, GNL could predict neutralization patterns for future viral sequences based on prior sequence and neutralization data (**Fig 7**). GNL was also capable of "transfer learning," using antibody-virus neutralization data from the CATNAP database to predict neutralization values for viruses in the CH505 data set, despite the absence of these sequences in CATNAP data (**Fig 8**). These capabilities could be used to inform antibody-based treatments and to aid in understanding virus evolution to escape antibodies [26,27,38–41].

Our approach addresses several important challenges in neutralization prediction. GNL effectively learns from large but sparse neutralization data sets such as CATNAP, where neutralization values for only around 4.5% of all antibody-virus pairs are measured. As data sets grow larger, they are also almost certain to increase in sparsity. Sharing information among antibodies with similar neutralization profiles is especially beneficial for antibodies with limited training data. Partitioning antibodies into groups also improves the computational efficiency of GNL. Combining sequence-based predictors

with low-rank matrix refinement allows us to incorporate both specific sequence features associated with antibody sensitivity or resistance and general patterns of neutralization across antibodies.

There are also several limitations of GNL that should be considered. First, as with any sequence-based prediction approach, our method is limited by the variants observed in the training data; rare or unobserved variants, which can be clinically important drivers of escape, might not be explicitly captured. It is also important to note that the IC50 values that we estimate are based on in vitro data sets using target cells that artificially express high levels of CD4 receptor. Thus, this measurement does not reflect in vivo potency directly, but rather an indirect, relative measure of neutralization susceptibility [30,42]. Absolute IC50 values therefore should not be interpreted as direct predictors of in vivo efficacy, and the relationship between in vitro neutralization and in vivo protection may vary across antibodies. Our approach is intended to compare relative sensitivity across viral variants and antibodies within the same experimental framework, rather than to infer absolute in vivo potency. Related to this point, our approach does not model virus sequence evolution, and thus GNL alone would not be sufficient to predict which antibody resistance mutations are most likely to evolve in an individual.

Our approach also involves several assumptions or technical limitations. We have assumed a linear relationship between viral sequence features and (log) neutralization values, which may not capture more complex nonlinear mutation effects. Similarly, our dimensional reduction approach uses principal components of viral sequence covariance, which may not capture the most relevant features for predicting neutralization. However, this is not an inherent limitation of GNL, which could be adapted to use arbitrary sequence features for prediction. Finally, the similarity metric that we use for grouping antibodies depends on overlapping neutralization measurements, which may be more limited for newly discovered antibodies or those tested against different panels of viruses.

GNL could be extended in multiple ways in future work. Our framework could apply the Bliss-Hill approach to estimate neutralization values for combinations of antibodies, provided that concentration-dependent neutralization values are available for multiple viruses and antibodies [37,43]. We could further extend our model to consider more complex relationships between viral genotype and neutralization (for example, ones including epistasis [44–48]), which could improve predictive power. The interpretability of sequence features could be enhanced by incorporating biologically meaningful representations such as sequence motifs [49,50] or known resistance mutations, rather than relying solely on principal component analyses. While we have focused on HIV-1 in this work, our framework could readily be extended to other rapidly evolving pathogens where neutralization data and genetic sequences are available.

## Supporting information

**S1 Text. Supplementary methods.**
(PDF)

**S1 Table. Summary of the data sources used in this study.**
(PDF)

**S2 Table. Model hyperparameters.** Summary of key parameters, their meaning, suggested values, and guidelines for adjustment based on data characteristics.
(PDF)

**S3 Table. Mutations that are shared across more than 10% of analyzed antibodies and ranked top 0.05%.** This table summarizes key mutations identified based on their contributions to the predicted log IC50 values. Mutations are selected from the top 0.05% according to the absolute values of their weight parameters, which quantify their contributions to the predicted log IC50, across all antibodies within the same antibody class. For these top-ranked mutations, the table reports the average contribution values and the number of antibodies in which each mutation appears within the top 0.05% (third column). To consider common patterns across antibodies of the same class, we further retained only

mutations shared by at least more than 10% of total antibodies within the top 0.05%. The numbers of preselected mutations in the top 0.05% are 455, 463, 463, 124, and 269 for V3-glycan, CD4bs, V2-apex, MPER, and Interface/FP bnAbs, respectively.
(PDF)

**S4 Table. Mean Pearson's R values based on cross-validation.** The mean cross-validation-based Pearson's R (CV-R) values for each antibody type are listed in the table, with individual CV-R values shown in Fig 6 of the main text. The mean values were calculated only for bnAbs with sufficient data to allow SLAPNAP to run. For V1V2, V3, CD4bs, and MPER bnAbs, the mean CV-R values of GNL are often more than 10% higher than those of other methods. The overall mean CV-R, averaged across antibody types, indicates that GNL achieves the highest accuracy, with approximately a 10% improvement.
(PDF)

**S5 Table. Antibodies from intrahost HIV-1 evolution data [27].** Antibodies marked with an asterisk (*) are classified differently in the literature. While earlier studies did not classify them as bnAbs [27,51], more recent research considers them to be bnAbs [52,53].
(PDF)

**S1 Fig. Overview of the neutralization imputation processes.** (a) The partially observed neutralization matrix (columns: viruses, rows: antibodies) is used as input to impute the missing elements (gray scale) of the neutralization matrix. (b) Schematic representation of grouping antibodies based on the similarity of their neutralization values. Within the same group, neutralization activities are more similar than those between different groups. (c) Temporarily complete missing values using models that learn the relationship between viral genetic data and neutralization values. We employ the novel grouped learning method, training the model to simultaneously learn the relationships between neutralization values and viral genetic sequences for a group of antibodies that share similar neutralization profiles (S1 Text). Here, to effectively reduce the number of model parameters and avoid the common over-parametrization issue, we applied a standard dimensional reduction technique to viral sequences and trained the model using the projected sequences. (d) Obtain the complete neutralization matrix, derived from the low-rank matrix approximation of the temporarily completed matrix from step c. The optimal matrix rank was determined based on the distribution of eigenvalues (S1 Text).
(TIFF)

**S2 Fig. Distribution of HIV sequences across multiple subtypes in the first and second principal modes of the genetic covariance matrix.** Each point in the cloud represents a viral sequence projected onto the principal modes of the covariance matrix derived from the one-hot encoded sequences. These principal modes yield projected sequences that are also used to learn the neutralization values. In this visualization, we present only subtypes that include more than 30 viral sequences that are used in our training data. Subtypes such as B, C, A, and AE are mapped to distinct regions in the space.
(TIF)

**S3 Fig. Variance explained and correlation between predicted and true neutralization values as a function of rank.** As a typical example, we set the fraction of observed data as 80%, and the validation values were withheld uniformly at random from the CATNAP dataset. (a) Eigenvalues for the filled neutralization matrices in the GNL and Einav et al. methods. The intermediate ranks of eigenvalues roughly follow a power law. Dashed lines show the power law fit with exponents of −3/7 and −5/7, respectively. (b) Profiles of the variance explained for GNL and Einav et al. methods. The "optimal" matrix rank values, defined as the minimum rank at which the explained variance exceeds 95%, are 20 and 45 for the GNL and Einav et al. methods, respectively. (c) Pearson's R between true and predicted neutralization values

shown as a function of rank. Including additional eigenmodes ultimately reduces predictive power. The rate of decrease in Pearson's R using the GNL method is more gradual than that of Einav et al., and it maintains higher values.
(TIF)

**S4 Fig. (a-c) MSE values across seven data observation fractions (5% to 90%) and six classes of the standard deviation of neutralization measures, for the Einav et al. method, Independent Neutralization Learning (INL), and the GNL model.** (d, e) Differences in MSE values between GNL and the other methods. GNL outperforms INL for antibodies with higher variability in neutralization and for lower fractions of observed data (e), suggesting that antibody grouping mechanisms aid in learning from variable cases. (f-h) MSE values across different numbers of training measurements and (i, j) their corresponding MSE differences. The grouping method benefits antibodies with fewer observed measurements, suggesting the advantage of grouping for sparsely observed antibody cases (j).
(TIF)

**S5 Fig. Matrix-rank dependence of prediction accuracy for IC80.** The same experimental conditions as in Fig 3 of the main text are used, with IC80 considered as an alternative viral sensitivity measure. Multiple accuracy metrics are used: (a) Pearson's $R$, (b) Spearman's $\rho$, (c) regression slope, (d) concordance correlation coefficients, and (e) MSE. Solid and dashed lines represent results from models trained with both sequence alignments and alignment-free features (AFFs), and with sequence alignments alone, respectively.
(TIF)

**S6 Fig. Matrix-rank dependence of prediction accuracy for Hill slope.**
(TIF)

**S7 Fig. Matrix-rank dependence of prediction accuracy for Instantaneous Inhibitory Potential (IIP).**
(TIF)

**S8 Fig. Prediction accuracy measured by the regression slope for multiple viral sensitivity measures.**
(TIF)

**S9 Fig. Prediction accuracy measured by the concordance correlation coefficient (CCC) for multiple viral sensitivity measures.** Multiple viral sensitivity measures are considered: (a) IC50, (b) IC80, (c) Hill slope, (d) IIP, which was evaluated at a fixed antibody concentration of $c = 1000$ $\mu$g/mL.
(TIF)

**S10 Fig. Impact of individual mutations on the log IC50 values for each class of antibodies.** The experimental setup is the same as in Fig 5 of the main text. Contributions to $\log(IC50)$ are obtained by averaging across multiple antibodies within the same antibody class. The numbers of antibodies included in this analysis are 110, 112, 112, 30, and 65 for (a) V3-glycan, (b) CD4bs, (c) V2-apex, (d) MPER, and (e) Interface/FP antibodies, respectively. Mutation sites are selected based on experimental studies [28,31–36].
(TIF)

**S11 Fig. Time to rebound in example simulations of ART interruption.** Mutations are selected by ranking the absolute values of their weight parameters, which represent contributions to the predicted log IC50 values, and retaining those within the top 0.05%. These mutations are then filtered to include only those present in more than 10% of antibodies within the same antibody class. (However, the points shown represent weight values across all antibodies, regardless of whether they fall within or outside the top 0.05% selection.) Detailed statistics and the effects of these mutations on antibody neutralization are summarized in S3 Table.
(TIF)

**S12 Fig. Performance comparison of GNL and existing methods based on Spearman's correlation coefficients.** The same experimental setup as in Fig 6 of the main text is used, with results shown for cross-validation–based Spearman's correlation.
(TIF)

**S13 Fig. Overall neutralization imputation performance for the individual host.** (a) Neutralization values are represented with purple, gray, and yellow colors, corresponding to observed, missing, and withheld values for validation, respectively. The withheld elements are chosen uniformly at random across antibodies and viruses. In this analysis, 80% of the total available data is observed. (b) A comparison of the true withheld neutralization values and the imputed values is shown on the x- and y-axes. Pearson's $R$, Spearman's $\rho$, MSE, and $p-$ values are, 0.89, 0.82, 9.42, and $10^{-90.9}$, respectively. (c) Accuracy Dependency on the Fraction of Observed Data. The overall accuracy of the GNL method is higher than that of the Einav et al. method (provided by Einav et al. [20]) as the fraction of observed data increases. (d) Dependency of accuracy on matrix rank $\rho$. The R values of the GNL method are consistently higher than those of the Einav et al. method across most rank values.
(TIFF)

## Author contributions

**Conceptualization:** Kai S. Shimagaki, John P Barton.

**Data curation:** Kai S. Shimagaki, Gargi Kher.

**Formal analysis:** Kai S. Shimagaki, John P Barton.

**Funding acquisition:** Rebecca M. Lynch, John P Barton.

**Investigation:** Kai S. Shimagaki, John P Barton.

**Methodology:** Kai S. Shimagaki, John P Barton.

**Project administration:** Kai S. Shimagaki, John P Barton.

**Resources:** Kai S. Shimagaki, John P Barton.

**Software:** Kai S. Shimagaki.

**Supervision:** John P Barton.

**Validation:** Kai S. Shimagaki, John P Barton.

**Visualization:** Kai S. Shimagaki.

**Writing – original draft:** Kai S. Shimagaki, John P Barton.

**Writing – review & editing:** Kai S. Shimagaki, Gargi Kher, Rebecca M. Lynch, John P Barton.

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
