## [Decision Letter · Decision Letter 0]

12 Dec 2025

PCOMPBIOL-D-25-01851

Predicting viral sensitivity to antibodies using genetic sequences and antibody similarities

PLOS Computational Biology

Dear Dr. Barton,

Thank you for submitting your manuscript to PLOS Computational Biology. After careful consideration, we feel that it has merit but does not fully meet PLOS Computational Biology's publication criteria as it currently stands. Therefore, we invite you to submit a revised version of the manuscript that addresses the points raised during the review process.

We look forward to receiving your revised manuscript.

Kind regards,

Adam Ewing

Academic Editor

PLOS Computational Biology

Amber Smith

Section Editor

PLOS Computational Biology

**Journal Requirements:**

At this stage, the following Authors/Authors require contributions: Kai Shimagaki, Gargi Kher, Rebecca M. Lynch, and John P. Barton. Please ensure that the full contributions of each author are acknowledged in the "Add/Edit/Remove Authors" section of our submission form.

5) We notice that your supplementary Figures, and Tables are included in the manuscript file. Please remove them and upload them with the file type 'Supporting Information'. Please ensure that each Supporting Information file has a legend listed in the manuscript after the references list.

**Reviewers' comments:**

Reviewer's Responses to Questions

**Comments to the Authors:**

Reviewer #1: Shimagaki et al. have developed a computational approach to predict log(IC50) for anti-HIV antibodies based on the semi-large existing data sets for virus/mab pairs. This is an admirable goal and could be valuable to a range of prevention/therapy/and cure studies for HIV and beyond. The work is well validated against existing approaches and appears substantially better.

A few bigger picture comments:

A main stated motivation for this work is that neutralization assays are hard or not available for many bnAb virus combinations. But it might be worth it to mention the type of sequencing that is required to generate data for this method and how much easier/cheaper that is to give a reader a concrete demonstration of how useful this would be and how this would fit into a clinical study (for instance, in participant inclusion criterion after a quick sequencing step before an inteventional study?)

It should be acknowledged that even if prediction is possible, unknown unknown HIV variants are practically problematic. Whether in circulation or within host such variants are potentially more relevant than more abundant variants -- historic work (Lorenzi et al) have shown abundant clonotypes are paradoxically less likely to drive relevant dynamics

It might be useful to include further notions of sensitivity from CATNAP (i.e. hill slope), since you have those data. This would also enhance the novelty beyond the Einav et al paper. One choice could be IIP at a relevant dose level, perhaps 1000µg/mL or something? You can find equation here (https://www.nature.com/articles/s41467-023-43384-y) and we found IIP correlating with viral load in HIV prevention studies with VRC01, also you might cite Laskey and Siliciano etc if you use that measure.

minor:

-"antibody-virus neutralization values" -- it would be helpful to be more precise in your initial description of the data, probably clarify catnap contains summary measures taken from in vitro TZMbl assays, and then define your measure of "sensitivity" as IC50, a summary statistic based on modeling titration curves

-A large caveat we also addressed in that paper mentioned above is that the in vitro IC50 is far lower than the observed in vivo potency (ie bnAbs work much worse in vivo than they appear to in vitro) which might also be good to note, I unfortunately don't know at all if this scaling factor should be the same between bnAbs

fig1a) I guess you don't need to redo all your work but from an interpretability pov it would be nice to show log10(IC50) so someone can naturally see 0.1, 1, 10 etc

fig1c) isn't predicted usually on the x-axis for this type of diagram?

-is R really the right measure to assess prediction score? perhaps a concordance correlation coefficient and showing y=x and the slope of the best line may be more relevant and illuminating for bias

fig3) what are the error bars? should be mentioned in legend

-"bnAbs" doesn't show up until page 6, not clear throughout which are bnAbs and which aren't

-did you ever explore whether certain bnAbs neutraliation is better predicted by looking only at partial sequences relevant for that binding site?

-I wondered how much error was introduced due to "resistance" saturation level. It might be useful to set all the saturated IC50s to be the same (e.g. 50µg/mL) and not worry about 20 vs 50 etc in your predictions

-refs have lots of (bibtex?) lower-case errors

Reviewer #2: This is a well-written manuscript that advances the problem of computational prediction of antibody neutralization of pathogens. The models are well-reasoned and rigorously tested, and comparisons to other state-of-the-art methods are fairly handled.

Major comments:

1) It is impressive that encoding viral sequences that the authors employ can predict held-out data well. However, we know from structure-function studies several critical motifs required for neutralization of specific bNAbs, and it is not clear if the reduced vector space for viral sequence encoding is sensitive enough to such mutations. For example, V3 glycan bNAbs typically require N332-glycan (not all, but most), and V2 apex bNAbs can be knocked-out by negatively charged amino acids at sites 169 and/or 171. Have the authors explored datasets of such mutants to test how well or not their models can predict the impact of these resistance mutations? Such knockout mutations are also important in the clinical setting of passively transferred bNAb(s) to viremic participants – where such mutations can be selected for in vivo. I strongly recommend the authors to explore such clinical datasets to test if their methods can predict such resistance development due to a few Env mutations that were strongly selected for by the passively transferred bNAbs. If successful, this will also demonstrate the applicability of the authors’ algorithm to this clinically relevant setting.

2) I recommend the authors to explore two potential avenues for biologically relevant characterization of viral sequence features, and test if these might improve the training-data fits and ultimately the prediction accuracy of their models:

i) Are the hypervariable loops of HIV Env treated any differently than the rest of the Env? It is well-known that these loops evolve rapidly by insertions/deletions in addition to point mutations; and this length and sequence variation implies that they cannot be meaningfully aligned. Therefore if the alignments were done differently in these regions, could they impact the encoding of the viral space which could then impact the neutralization predictions? Bette Korber et al. introduced alignment-free characteristics of length, net charge and total number of glycans to characterize these hypervariable loops and showed that each bNAb class was significantly impacted by one or more of these features (https://pubmed.ncbi.nlm.nih.gov/30629920/). It is not clear to me if use of numeric variables is amenable to the encoding strategy of the authors, but if possible, this might be interesting to test to see if training fits and prediction accuracy improve.

ii) In the “one-hot” encoding of viral sequences, the authors use an alphabet of 21 letters (20 amino acids and a gap). This alphabet does not include potential N-linked glycan sites (N-X-S or N-X-T, where X is any aa except Pro, PNGS). Virtually each known HIV bNAb has to interact with glycans, and often glycan gain/loss mutations (e.g. 332-NIS-334 to 332-NIN-334 for V3 glycan bNAbs) are important, even if the parental Asn site is not mutating. In contrast to above, the addition of one more letter in the alphabet (‘O’ for Asn in PNGS) should be straightforward to add to the already available encodings.

Minor comments:

1) I recommend the authors to report significance using statistical tests (e.g. non-parametric tests) while reporting correlation coefficients between observed and predicted neutralization titers.

2) I could not readily find the names of the antibodies used for example in the CH505 dataset. I urge the authors to provide summary tables in the supplement to quickly guide the readers about the makeup of the datasets used.

Reviewer #3: My review is uploaded.

**Have the authors made all data and (if applicable) computational code underlying the findings in their manuscript fully available?**

The PLOS Data policy requires authors to make all data and code underlying the findings described in their manuscript fully available without restriction, with rare exception (please refer to the Data Availability Statement in the manuscript PDF file). The data and code should be provided as part of the manuscript or its supporting information, or deposited to a public repository. For example, in addition to summary statistics, the data points behind means, medians and variance measures should be available. If there are restrictions on publicly sharing data or code —e.g. participant privacy or use of data from a third party—those must be specified.requires authors to make all data and code underlying the findings described in their manuscript fully available without restriction, with rare exception (please refer to the Data Availability Statement in the manuscript PDF file). The data and code should be provided as part of the manuscript or its supporting information, or deposited to a public repository. For example, in addition to summary statistics, the data points behind means, medians and variance measures should be available. If there are restrictions on publicly sharing data or code —e.g. participant privacy or use of data from a third party—those must be specified.

Reviewer #1: Yes

Reviewer #2: Yes

Reviewer #3: Yes

PLOS authors have the option to publish the peer review history of their article (what does this mean? ). If published, this will include your full peer review and any attached files.). If published, this will include your full peer review and any attached files.

**Do you want your identity to be public for this peer review?** For information about this choice, including consent withdrawal, please see our For information about this choice, including consent withdrawal, please see our Privacy Policy ..

Reviewer #1: **Yes:** Daniel B ReevesDaniel B Reeves

Reviewer #2: No

Reviewer #3: No

**Figure resubmission:**
---

## [Decision Letter · Decision Letter 1]

5 Mar 2026

Dear Dr Barton,

We are pleased to inform you that your manuscript 'Predicting viral sensitivity to antibodies using genetic sequences and antibody similarities' has been provisionally accepted for publication in PLOS Computational Biology.

Best regards,

Adam Ewing

Academic Editor

PLOS Computational Biology

Amber Smith

Section Editor

PLOS Computational Biology

Reviewer's Responses to Questions

**Comments to the Authors:**

Reviewer #1: Authors have done a comprehensive job addressing comments.

Reviewer #2: None.

**Have the authors made all data and (if applicable) computational code underlying the findings in their manuscript fully available?**

The PLOS Data policy requires authors to make all data and code underlying the findings described in their manuscript fully available without restriction, with rare exception (please refer to the Data Availability Statement in the manuscript PDF file). The data and code should be provided as part of the manuscript or its supporting information, or deposited to a public repository. For example, in addition to summary statistics, the data points behind means, medians and variance measures should be available. If there are restrictions on publicly sharing data or code —e.g. participant privacy or use of data from a third party—those must be specified.requires authors to make all data and code underlying the findings described in their manuscript fully available without restriction, with rare exception (please refer to the Data Availability Statement in the manuscript PDF file). The data and code should be provided as part of the manuscript or its supporting information, or deposited to a public repository. For example, in addition to summary statistics, the data points behind means, medians and variance measures should be available. If there are restrictions on publicly sharing data or code —e.g. participant privacy or use of data from a third party—those must be specified.

Reviewer #1: None

Reviewer #2: Yes

PLOS authors have the option to publish the peer review history of their article (what does this mean? ). If published, this will include your full peer review and any attached files.). If published, this will include your full peer review and any attached files.

**Do you want your identity to be public for this peer review?** For information about this choice, including consent withdrawal, please see our For information about this choice, including consent withdrawal, please see our Privacy Policy ..

Reviewer #1: **Yes:** Daniel B ReevesDaniel B Reeves

Reviewer #2: No

---

## [Editor Report · Acceptance letter]

PCOMPBIOL-D-25-01851R1

Predicting viral sensitivity to antibodies using genetic sequences and antibody similarities

Dear Dr Barton,

I am pleased to inform you that your manuscript has been formally accepted for publication in PLOS Computational Biology. Your manuscript is now with our production department and you will be notified of the publication date in due course.

With kind regards,

Zsofia Freund
